# Electronic Patient-Reported Outcome Measures (ePROMs) Improve the Assessment of Underrated Physical and Psychological Symptom Burden among Oncological Inpatients

**DOI:** 10.3390/cancers15113029

**Published:** 2023-06-01

**Authors:** Eva Warnecke, Maria Rosa Salvador Comino, Dilara Kocol, Bernadette Hosters, Marcel Wiesweg, Sebastian Bauer, Anja Welt, Anna Heinzelmann, Sandy Müller, Martin Schuler, Martin Teufel, Mitra Tewes

**Affiliations:** 1Department of Palliative Medicine, West German Cancer Center, University Hospital Essen, University of Duisburg-Essen, 45122 Essen, Germany; eva.warnecke@uk-essen.de (E.W.);; 2Department of Psychosomatic Medicine and Psychotherapy, West German Cancer Center, LVR-Klinikum Essen, University of Duisburg-Essen, 45122 Essen, Germany; 3Directorate of Nursing, University Hospital Essen, 45122 Essen, Germany; 4Department of Medical Oncology, West German Cancer Center, University Hospital Essen, University of Duisburg-Essen, 45122 Essen, Germany

**Keywords:** ePROMs, nurse-reported, symptom burden

## Abstract

**Simple Summary:**

Professionals of the healthcare system face the challenge of providing suitable care delivery for inpatients with advanced cancer. External assessments, such as nurse-reported symptom burden, tend to underrate patients’ distress, as has already been shown in different studies. In contrast to that, patient-reported outcome measures (PROMs) have been shown to be effective as a systematic assessment; however, they are not yet implemented into daily routine. The aim of this retrospective study is to compare the information provided by PROMs and nurse-reported assessment to identify overlaps and differences in the current symptom burden. Therefore, collected data were analyzed from both PROMs and nurse-reported assessment from 230 inpatients of a major German Comprehensive Cancer Center. We discovered that some features of physical and psychological distress were underrated by nursing staff. Supplementing daily symptom assessment used by nursing staff with systematic ePROMs may improve the quality of supportive and palliative care of inpatients with advanced cancer.

**Abstract:**

For advanced cancer inpatients, the established standard for gathering information about symptom burden involves a daily assessment by nursing staff using validated assessments. In contrast, a systematic assessment of patient-reported outcome measures (PROMs) is required, but it is not yet systematically implemented. We hypothesized that current practice results in underrating the severity of patients’ symptom burden. To explore this hypothesis, we have established systematic electronic PROMs (ePROMs) using validated instruments at a major German Comprehensive Cancer Center. In this retrospective, non-interventional study, lasting from September 2021 to February 2022, we analyzed collected data from 230 inpatients. Symptom burden obtained by nursing staff was compared to the data acquired by ePROMs. Differences were detected by performing descriptive analyses, Chi-Square tests, Fisher’s exact, Phi-correlation, Wilcoxon tests, and Cohen’s r. Our analyses pointed out that pain and anxiety especially were significantly underrated by nursing staff. Nursing staff ranked these symptoms as non-existent, whereas patients stated at least mild symptom burden (pain: mean_NRS/epaAC_ = 0 (no); mean_ePROM_ = 1 (mild); *p* < 0.05; r = 0.46; anxiety: mean_epaAC_ = 0 (no); mean_ePROM_ = 1 (mild); *p* < 0.05; r = 0.48). In conclusion, supplementing routine symptom assessment used daily by nursing staff with the systematic, e-health-enabled acquisition of PROMs may improve the quality of supportive and palliative care.

## 1. Introduction

As part of routine clinical practice in the inpatient setting with advanced cancer patients (ACPs), medical and nursing staff face the daily challenge of providing suitable care delivery [1]. Since ACPs are often confronted with physical and psychological symptoms, symptom management has gained increasing importance in providing individualized and high-quality patient-centered care [2]. Therefore, the expertise of medical and nursing staff in the inpatient and outpatient settings plays an important role. Additionally, the patient’s declaration of the severity of symptom burden influences the procedure of decision making. To prioritize patients’ symptom burden, one of the key questions is what bothers them the most. Hence, there is a growing interest in using validated assessments. In most inpatient settings, validated assessments are the standard sources of reporting symptoms. They are often and firstly used by nursing staff [3]. However, these assessments often are the only source of information on symptom burden, even though different studies have already shown that healthcare professionals have a tendency to underrate the severity of symptoms compared to the patients themselves and their relatives [4]. In addition, other studies have estimated that the severity of ACPs’ symptom burden often remains undetected and unaddressed [5]. Consequently, a considerable number of patients suffer from unaddressed extensive physical and psychological symptom burden [6]. 

To improve the circumstances of high and undetected symptom burden, in Germany, the German S3-guidelines on palliative care and the German network for quality development (DNQP), among others, recommend the incorporation of patient-reported outcome measures (PROMs) in clinical routine. PROMs “describe any report or measure of health reported by the patient, without external interpretation by a clinician or researcher” [7]. One of the overarching goals according to the German S3-guidelines is the “improvement of symptom control in palliative care for patients with incurable cancer and their families” [8]. The systematic use of PROMs could help us to achieve this target. Patients using PROMs declare inapparent symptoms and high symptom burden earlier [3]. Additionally, some versions of ePROMs have the feature of directly informing nursing staff about high symptom burden. As a result, nursing staff can take a direct response and initiate adequate symptom management immediately [9]. Since the use of PROMs facilitates the earlier detection of symptoms and needs, it may lead to the earlier integration of palliative care than before. 

Studies have shown that integrating palliative care at an early stage of cancer’s disease leads to significant improvements in quality of life and mood. Comparing ACPs receiving standard care with those who are receiving early palliative care, it becomes evident that they had less aggressive care at the end of life [10,11]. Moreover, the systematic use of PROMs leads to an improvement in communication between patients and healthcare providers [12,13], increased survival rates [14,15,16], and less visits to the emergency room [9]. 

To our knowledge, there are studies that have dealt with PROMs to identify patients with unmet palliative care needs [17], its integration in clinical settings [18,19], the effects of the early integration of palliative care [11], or symptom burden [20,21]. Moreover, a few studies have compared PROMs and clinician-reported symptom burden [22]. 

However, our study is the first to directly compare the effects of and differences in using different tools (self-reported and nurse-reported) simultaneously to gather information about the severity of symptom burden. 

On the one hand, we are accustomed to the information provided by the established assessments used by nursing staff. On the other hand, the incorporation of PROMs is required to be part of the daily routine, but it is still not properly integrated into German hospitals. 

The purpose of this study is to compare the information provided by ePROMs with nurse-reported symptom burden. We aim to identify overlaps and differences in current symptom burden. Our objective is to determine whether symptom burden reported solely by nurses, as established in daily routine, reflects the patient’s perspective and their subjective severity of symptoms. Furthermore, we wish to decipher a better understanding of the value of information provided by PROMs. Additionally, based on our findings, we will derive advantages and disadvantages that may influence our attitude towards the ideal use of self- and nurse-reported tools. 

## 2. Materials and Methods

We designed a retrospective, non-interventional, monocentric study, developed among the inpatients of a major German Comprehensive Cancer Center. A total of 230 inpatients with a histologically confirmed tumor disease were recruited between September 2021 and February 2022. 

### 2.1. Study Protocol, Patient Recruitment

Patients who met the eligibility criteria, defined as above 18 years of age, undergoing inpatient treatment, and having a histologically confirmed tumor disease were asked to report their symptom burden once during their inpatient treatment. They were requested to do so using our electronic version of PROMs (ePROMs) called ePOS, an abbreviation of electronic psycho-oncological and palliative screening. We considered any patient who received our version of PROMs on the day of the inpatient admission or within the first days of the inpatient stay as a participant in our study. The timing of recruitment depended on the patients’ general condition. Due to organizational constraints, such as a limited number of tablets available for each inpatient ward, patients were instructed to fill in PROMs immediately after receiving them. Reading the instructions and completing the information typically takes about 10 to 15 min. During patient recruitment, we also included patients with repeated inpatients stays. 

### 2.2. Questionnaires

The nursing staff of the medical oncology wards at the observed University Hospital regularly assess the patients’ clinical condition using an established tool. Currently, they use an assessment called the outcome-oriented nursing assessment instrument AcuteCare (epaAC 2.1) [23], which consists of more than 60 validated items. Patient information is collected digitally, allowing real-time data usage. The information is available in the hospital’s computerized system within minutes, making it accessible to all clinicians involved in the patients’ care. The assessment focuses on identifying nursing-relevant risk factors and evaluating patients’ abilities in various categories such as activity, mobility, grooming, nutrition, elimination, cognition and consciousness, communication and interaction, sleep, breathing, pain, and wounds. In addition, other important aspects are documented, including pain, recorded with validated tools such as the numeric rating scale (NRS) [24] or the German version of pain assessment in advanced dementia (PAINAID), abbreviated as BESD [25]. Furthermore, the risk for developing pneumonia, current nutritional status, and mucositis is also documented. Nursing staff compile this information once per shift, deriving it from observing and assessing patients’ behavior, body language, and descriptions during their interactions and visits.

The ePROMs questionnaire used in our study, called ePOS (electronic psycho-oncological and palliative screening), consists of 51 items. It includes a total of four questionnaires and two questions regarding the patient’s subjective need for psychological or palliative care support. The first questionnaire included is the Patient Health Questionnaire 8 (PHQ-8), which contains eight questions. It has been “established as a valid diagnostic and severity measure for depressive disorders in large clinical studies” [26]. The second questionnaire integrated into our version of ePROMs is the Generalized Anxiety Disorder GAD-7. “Both the GAD-7 and its abbreviated two-item version (GAD-2) have good operating characteristics for detecting generalized anxiety, panic, social anxiety and post-traumatic stress disorder” [27]. In addition, ePOS includes the Hornheider Screening Instrument (HSI), which is used to identify patients with “psychosocial care needs” [28]. To collect information about symptom burden, ePOS utilizes the minimal documentation system MIDOS2, which is “a German version of the Edmonton Symptom Assessment Scale (ESAS)” [29]. It contains questions that gather information about the current state of various typical symptoms such as “nausea, dyspnoea [sic], constipation, weakness, tiredness, anxiety” [29]. Symptoms are ranked on a four-point Likert scale (0 = no symptoms, 1 = mild symptoms, 2 = moderate symptoms, 3 = severe symptoms) (Table 1).

### 2.3. Statistical Methods

The sample size calculation was conducted using G*Power version 3.1.9.6 [30]. We aimed to achieve a power of 95% and an α-error of 5% for our analyses by defining the appropriate test families.

Data management and analyses were performed using the Statistical Program for Social Sciences SPSS version 27.0 and 29.0.0.0 (IBM Corporation, Armonk, NY, USA). Descriptive data, including median, mean values, and z-standardization, were generated to provide an overview of the patients’ characteristics and to identify differences between ePROMs and nurse-reported symptom burden. As some items in ePROMs and nurse-reported symptom burden were ranked using different scale levels, we aimed to create comparability downscaling some scales, from a four-point Likert scale to a nominal type (yes/no). We acknowledge that the process of downscaling may result in a potential loss of information. The original four-point Likert scale allows responses to encompass finer distinctions and nuances. By converting it to a binary scale, the differentiation between different levels of symptom burden is eliminated [31]. 

We conducted a three-step approach to compare the severity of symptom burden reported using the aforementioned tools. 

Due to the discrepancy that ePROMs information was collected only once while information in epaAC was obtained daily, we decided to analyze the data provided by epaAC from the same day and, if possible, from nearly the same time that ePROMs were filled in. This approach aimed to ensure a valid basis for further analyses. Additionally, we analyzed every PROM and its corresponding entry in epaAC, understanding that it might introduce bias. However, we proceeded in this manner to derive advantages and disadvantages from our results. 

First, we focused on the results of the descriptive analysis, comparing median or mean values and z-standardization based on the type of scale level. Since collected data were not normally distributed, non-parametric tests were used for further analyses. Different tests were performed to assess significant differences in symptom burden, considering the variety of scale levels. For scales rated as nominal, the Chi-squared test (X^2^) was performed. The null hypothesis was defined as no differences between ePROMs and nurse-reported symptom burden. For items with expected cell frequencies less than five, such as nausea, dyspnea, depressiveness, and concentration, Fisher’s exact (*p*) was conducted [32].

The Phi-coefficient (φ) was also calculated for items with a rejected null hypothesis as a measure of association with effect sizes ranging from 0.1 to 0.3 (small), 0.3 to 0.5 (medium), and greater than 0.5 (large) [33].

In addition, we performed the Wilcoxon test with a defined significance level of *p* < 0.05 for the symptom burden of pain and anxiety, which were rated on a four-point Likert scale. As our sample size consisted of more than 25 participants, we focused on the asymptotic significance of the Wilcoxon test as an indicator of significant difference, following the work of Harris and Hardin [34]. 

To measure the strength of the relationship between two symptoms analyzed with the Wilcoxon test, we conducted Cohen’s r as the effect size. The effect sizes of Cohen’s r are the same as those for the Phi-coefficient (φ): 0.1–0.3 (small), 0.3–0.5 (medium), and greater than 0.5 (large) [35].

## 3. Results

The sample size calculation recommended recruiting the highest number of participants for the X^2^ test family. Based on this recommendation, a total of 230 participants were recruited for the study. Data collection took place between September 2021 and February 2022. Eligible patients received our digital version of PROMs called ePOS once during their inpatient treatment. Some patients with repeated inpatient stays were asked to fill in PROMs more than once.

Out of the total collective of participants, 110 (47.8%) patients completed the tablet-based questionnaire in its entirety. Additionally, 189 (82.2%) patients started filling in their information at least once. Among them, 34 (14.8%) patients filled in the information twice, while 7 (3.0%) patients self-reported their symptoms three times. Unfortunately, during the study period, 13 (5.7%) patients, who reported their symptom burden at least once using our version of ePROMs, passed away.

### 3.1. Patient Characteristics

The main characteristics of the inpatients who received ePROMs are as follows (Table 2):

The analyses of sociodemographic features revealed that the majority of participants were female (*n* = 127; 55.2%). The mean age was 56 years (SD ± 15.07), ranged from 18 to 86 years. Out of the total participants, 111 (48.3%) were married, and 119 (51.7%) had children. Although most patients suffered from an incurable disease (UICC IV) (*n* = 123; 53.5%), 72 (31.3%) patients were still employed. 

The analyses of medical features delivered the following results. The leading tumor diagnosis was soft-tissue sarcoma (*n* = 75; 39.1%), followed by lung cancer (*n* = 38; 20.1%). As already mentioned, some participants received ePROMs more than once because of repeated inpatient treatment. Note that the number of tumor diagnoses differs from the total sample size of 230. It was counted only once to avoid confounding due to the fact of it being an unchangeable feature.

At the time of presentation, 175 (76.1%) were diagnosed with metastatic disease. Among the participants with an advanced tumor disease at stage III and IV UICC (initial), the majority (*n* = 178; 77.4%) had a good ECOG performance status (ECOG 0–2; *n* = 150; 65.2%). Additionally, by the time of screening, most of the participants had received first-line therapy (*n* = 146; 63.5%), including chemotherapy, radiation, and/or immune/targeted therapies. 

### 3.2. Differences in Symptom Burden

The comparison of descriptive data, including median or mean values and z-standardization, highlights differences in symptom burden between nurse-reported symptoms and ePROMs. The analyzed categories, such as nausea, dyspnea, difficulties with sleep, depressiveness/sadness, weakness/exhaustion, concentration/attention, and pain and anxiety, showed varying ratings between the two sources. 

The analysis of the data indicates that symptoms, particularly those that may not be apparent at first sight and are associated with psychological distress, appear to be underrated by nursing staff. Patients using ePROMs reported feeling burdened by difficulties with sleep, depressiveness, weakness, and concentration, while nursing staff rated these features as non-existent (Table 3). Furthermore, the analysis of pain and anxiety also showed differences between ePROMs and nurse-reported symptom burden. Patients using ePROMs declared that they experience mild symptoms of pain and anxiety, whereas nurses tended to rate these symptoms as non-existent (Table 3). 

However, it is worth noting that the symptoms of nausea and dyspnea/breathing were rated similarly by both ePROMs and nurse-reported assessment tools, with both sources indicating an inexistent burden (Table 3).

Statistical tests were conducted to further examine the differences in symptom burden. The Chi-squared (X^2^) and the Fisher’s exact (*p*) tests indicated that only difficulties with sleep showed a statistically significant difference (*p* < 0.00, Table 3) with a small negative measure of association (φ = −0.22). 

Regarding pain and anxiety, the non-parametric Wilcoxon test showed a statistically significant difference (*p* < 0.05) between ePROMs and nurse-reported symptom burden. 

The effect sizes, measured by Cohen’s r, indicated a medium effect size for pain (r = 0.46 and r = 0.46) and anxiety (r = 0.48). 

## 4. Discussion

In this retrospective, non-interventional, monocentric study, we compared the severity of symptom burden reported by nursing staff, using established and validated assessments, to symptom burden reported by patients themselves using ePROMs. Our findings indicate that ePROMs provide a different perspective on symptom burden compared to nurse-reported symptoms, particularly in the areas of psychological distress. These results emphasize the importance of incorporating PROMs into our daily routine. 

Our patient cohort suffered from relatively low symptom burden, which seemed to be independent of the method used for gathering information. Descriptive analyses have revealed that in most cases, symptoms were either non-existent or perceived as mild. 

In contrast to epaAC, that focuses on identifying nursing-relevant risk factors and evaluating patients’ abilities, our ePROMs primarily address features of psychological health (Table 4).

Although the overall burden appeared low, there were differences in the information collected by nurses and ePROMs. Our data showed that specific symptoms remained undetected or underrated by nursing staff, which is consistent with previous studies [14,36]. Symptoms associated with psychological distress [36], such as difficulties with sleep, pain, and anxiety, were underrated in the nurse-reported method, highlighting the potential of self-reported instruments in identifying patients with unmet clinical needs. These findings may help to explain why other studies have found that patients using PROMs tend to report their symptoms earlier [3] than they are apparent to nursing staff. It could be assumed that patients may feel uncomfortable directly discussing their symptoms and needs. Further investigations are needed to explore patients’ motivation for reporting symptoms earlier when using PROMs. 

Analyzing the data of every participant, regardless of whether ePROMs were entirely completed, we derived conclusions regarding the appropriateness of using ePROMs as a single source of information for every inpatient. The correct utilization of PROMs depends on various conditions, such as the patients’ current condition, disease progression, or cognitive function [37]. In our analysis, a high number of missing data indicated that many inpatients refused to complete ePROMs for various reasons, such as subjectively having no need for a screening, having already undergone screening, no change in symptom burden, or experiencing a reduced general condition. These results align with the findings from other studies, which identified that patients who feel too sick or do not feel inclined to complete PROMs tend to be non-compliant [3]. Furthermore, the inability of patients to use electronic devices can contribute to incomplete ePROMs [13]. Consequently, patients with a high symptom burden or those who are unfamiliar with electronic devices [38] may be underrepresented. Therefore, it is important to recognize that using PROMs as a single source of information may not be suitable for every subgroup as it can lead to a lack of proper clinical information. 

Importantly, utilizing a digital version of PROMs offers clear advantages. Real-time feedback [39,40] is a crucial feature, allowing clinicians to monitor patients’ condition and promptly respond when symptom burden is high [9]. As already stated by Eid et al., this is one of “the ePROs merits in oncology clinical care, which is their ability to enable regular and real-time monitoring of patient symptoms and needs, therefore enhancing the efficiency of care” [13]. It enables timely interventions and contributes to reducing overall symptom burden and improving patients’ quality of life [10]. Furthermore, it can reduce the number of ER visits [9]. These findings highlight the urgency of systematically integrating PROMs into German hospitals. 

Simultaneously using both assessments employed by nursing staff and ePROMs provides additional information regardless of the patients’ condition or compliance. The approach suggests that customized treatment for certain patients may only be ensured by using both tools complementarily at the same time. This finding underlines the work of Ahern et al. [40]. 

## 5. Limitations

This study has a number of limitations that should be acknowledged. First, it is a monocentric study. Additionally, it was conducted in a leading center for patients with soft-tissue sarcoma, leading to potential center bias. The choice of location and its feature as a center may limit the generalizability of the findings to other settings or patient populations. 

Second, our data may not include patients with high symptom burden or those who are in a somnolent or comatose state, as our version of ePROMs may not be suitable for patients with a poor general condition. This could result in the exclusion of a specific subset of patients, limiting the overall representativeness of the study. 

Third, as the study population had a relatively low symptom burden, the differences between nursing and ePROMs assessments may not be fully explored. Further research is needed to investigate the disparities between these assessments in inpatients with higher clinical burden.

Fourth, the availability of our version of ePROMs, only available in the German language, may exclude patients with foreign backgrounds who do not speak German. This could lead to the underrepresentation of foreign patients. 

Additionally, the analysis of PROMs can be challenging due to missing data, which may introduce bias and limit the accuracy of the results. Efforts should be made to address missing data issues and consider appropriate methods for handling missing data in future studies.

Furthermore, a significant limitation in data analysis is that ePROMs and the assessments used by nursing staff did not employ the same questionnaires and scales. This lack of standardization makes it difficult to analyze and directly compare each item. It results in a potential loss of information. To improve comparability, some four-point Likert scales had to be downscaled to a nominal type, which may further limit the precision of the analysis.

## 6. Conclusions

Overall, our study revealed discrepancies between ePROMs and nurse-reported symptom burden, particularly in the assessment of symptoms dealing with psychological distress. Symptoms, such as anxiety and difficulties with sleep were rated milder by nursing staff compared to patients using ePROMs. A similar result was observed for pain.

Based on these findings, it is recommended to utilize both ePROMs and assessments conducted by nursing staff in conjunction with each other. This combined approach enables a more comprehensive and accurate evaluation of symptom burden. By supplementing daily symptom assessments used by nursing staff with systematic ePROMs, healthcare providers can gain a more holistic understanding of patients’ symptoms regardless of patients’ compliance. Consequently, the quality of supportive care for ACPs in the inpatient setting can be improved.

However, further research is needed to fully understand the benefits and limitations of incorporating ePROMs in routine clinical practice.

## Figures and Tables

**Table 1 cancers-15-03029-t001:** Overview of the content of nurse-reported assessment and ePROMs.

Nurse-Reported	ePROMs (ePOS)
Basic data: BMI, level of care, home care services	Psycho-oncological and palliative care needs
Ambulating	Depression: Patient Health Questionnaire PHQ-8
Personal hygiene and dressing	Anxiety: Generalized Anxiety Disorder Scale GAD-7
Feeding	Psychosocial burden: Hornheider Screening Instrument HSI
Continence and toileting	Symptom burden: Minimal documentation system MIDOS2
Communication and interaction	
Sleep	
Regulation of body functions	
Pain and feelings	
Decubitus and wounds	
Wound documentation and special documentation including NRS for pain	

**Table 2 cancers-15-03029-t002:** Demographics and patients’ characteristics.

		**Total (*n* = 230)**
**Sociodemographic characteristics**		
Sex *n* (%)	Female	127 (55.2)
	Male	103 (44.8)
Age mean y (±SD)		55.99 (±15.07)
Marital status *n* (%)	Married In a relationship Single Divorced Living separately Widowed Missing information	111 (48.3) 18 (7.8) 32 (13.9) 9 (3.9) 2 (0.9) 9 (3.9) 49 (21.3)
Children *n* (%)	Yes	119 (51.7)
	No	59 (25.7)
	Missing information	52 (22.6)
Employment *n* (%)	Yes	72 (31.3)
	No	97 (42.2)
	Missing information	61 (26.5)
**Medical characteristics**		
Cancer diagnosis * *n* (%)	Soft-tissue sarcoma	75 (39.1)
	Lung cancer	38 (20.1)
	Uveal melanoma	21 (11.1)
	Gastrointestinal cancer	19 (10.1)
	Hepatobiliary cancer and pancreatic cancer	13 (6.9)
	Others	23 (11.0)
Line of therapy *n* (%)	0	5 (2.2)
	1	146 (63.5)
	2	39 (17.0)
	3	18 (7.8)
	>3	22 (9.6)
ECOG *n* (%)	ECOG 0	78 (33.9)
	ECOG 1	45 (19.6)
	ECOG 2	27 (11.7)
	ECOG 3	12 (5.2)
	ECOG 4	1 (0.4)
	Missing information	67 (29.1)
Initial tumor stage (UICC) *n* (%)	I	7 (3.0)
	II	17 (7.4)
	III	55 (23.9)
	IV	123 (53.5)
	Missing information	28 (12.2)
Metastasized *n* (%)	Yes	175 (76.1)
	No	54 (23.5)
	Missing information	1 (0.4)
Psychiatric treatment *n* (%)	At the moment	13 (5.7)
	Never	134 (58.3)
	Before	27 (11.7)
	Missing information	56 (24.3)
Tranquillizer, antidepressants, hypnotics *n* (%)	Daily use	20 (8.7)
	Sometimes	18 (7.8)
	Never	140 (60.9)
	Missing information	52 (22.6)

* unchangeable feature was counted only once during data collection (*n* ≠ 230) to avoid confounding.

**Table 3 cancers-15-03029-t003:** Overview of descriptive and further analysis.

**Nominal Ranked Symptoms, Results of the Chi-Squared Tests (X^2^), the Fisher’s Exact (*p*), and the Phi-Coefficient (φ)**							
**Symptom**	*n*	**Median**	**Symptom Burden**	**X^2^**	**Asymp. Sig.**	* **p** *	**φ**
Nausea (ePOS)	174	0.00	No (0)	4.35 ***	-	0.10	-
Nausea (epaAC)	225	0.00	No (0)				
Dyspnea (ePOS)	173	0.00	No (0)	1.36 ***	-	0.29	-
Breathing (epaAC)	225	0.00	No (0)				
Difficulties with sleep (ePOS)	164	1.00	Difficulties with sleep (1)	7.56	0.01(*n* = 161)	-	−0.22
Difficulties with sleep (epaAC)	225	0.00	No difficulties with sleep (0)				
Depressiveness (ePOS)	172	1.00	Depressiveness (1)	0.73 ***	-	1.00	-
Sadness (epaAC)	222	0.00	No sadness (0)				
Weakness (ePOS)	174	1.00	Weakness (1)	0.00	0.56 (*n* = 170)	-	-
Exhaustion (epaAC)	225	0.00	No exhaustion (0)				
Concentration (ePOS)	174	1.00	Difficulties with concentration (1)	0.81 ***	-	1.00	-
Attention (epaAC)	225	0.00	No difficulties with attention (0)				
**Symptoms Ranked with 4-point Likert, Results of the Wilcoxon Test, and the Cohen’s r**							
**Symptom**	** *n* **	**Mean**	**Symptom Burden**	**Z**	**Asymp. Sig.**	***p*-Value**	**r**
Pain (ePOS)	175	1.06	Mild pain (1)	−8.54 ^1^ −8.55 ^2^	0.00	0.00	0.46 ^1^ 0.46 ^2^
Pain (NRS) ^1^	226	0.25	No pain (0)				
Pain (epaAC) ^2^	224	0.21	No pain (0)				
Anxiety (ePOS)	174	0.01	No anxiety (0)	−8.83	0.00	0.00	0.48
Anxiety (epaAC)	222	0.93	Mild anxiety (1)				

Nominal rated symptoms: 0 = no; 1 = yes; *** expected cell frequencies less than five. Four-point Likert scale: 0 = no symptom burden; 1 = mild symptom burden; 2 = moderate symptom burden; 3 = severe symptom burden. Nurse-reported: epaAC and NRS. PROMs: ePOS. ^1^ pain (ePOS vs. NRS). ^2^ pain (ePOS vs. epaAC).

**Table 4 cancers-15-03029-t004:** Comparison of the advantages and disadvantages of ePROMs and nurse-reported symptom burden.

	ePROMs	Nurse-Reported
Advantages	Unrecorded stresses become transparent	Daily overview
	Subjective symptom burden in the spotlight	Independent of patients’ condition
	Psychological distress to be detected	
	Alerts when symptom burden is high	
Disadvantages	Depends on: - Compliance - Condition	Focused on nursing related risk factors and patients’ abilities
	Only once while inpatient undergoing treatment	Less focus on psychological distress

## Data Availability

The data presented in this study are available on request from the corresponding author. The data are not publicly available due to privacy and ethical considerations.

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
