# Peer review of "Electronic Patient-Reported Outcome Measures (ePROMs) Improve the Assessment of Underrated Physical and Psychological Symptom Burden among Oncological Inpatients"

_cancers, 2023, doi:10.3390/cancers15113029_

Round 1

Reviewer 1 Report

The Manuscript presents novel insights on the integration of electronic patient reported outcome measures (ePROM) to the nurse-reported symptom assessment when addressing physical and psychological symptoms in the context of advanced cancer inpatients and through a monocentric retrospective study on 230 patients.

The topic is of extreme interest, being the need to assess and address physical as well as psychological symptoms increasingly gaining consideration in the advanced cancer setting. I thank the Authors for the interesting reading.

The Introduction flows well and incorporates both the rationale of the study and the research question. Yet, I think it should be more detailed in some parts:

1. I think the Manuscript could benefit from a more detailed list of the advantages associated to the systematic use of PROMs, like symptom control, patient satisfaction with care, health-related quality of life, decreased hospitalization and emergency-department visits, improved overall survival (e.g., Barbera et al., 2020; Basch et al., 2018; Denis et al., 2019; Detmar et al., 2002; Velikova et al., 2004).

2. At a certain point Authors mention early palliative care. They say that “one of the overarching goals according to the German S3-guideline is the improvement of symptom control IN palliative care for patients with incurable cancer and their families”. At the same time, Authors say that “the early integration of palliative care is a target for ACP” and that “the use of PROM could help us to understand WHEN AND WHY OUR PATIENTS NEED PALLIATIVE CARE the most”. It is not clear which role Authors attribute to PROMs in the context of early palliative care. This is of great importance, because based on the early palliative care model, all patients should be referred to the palliative care within 8 weeks since the cancer diagnosis and not just based on the assessment of the severity of their symptoms (e.g., Bigi et al., 2023). Once on early palliative care, they will be monitored for their physical and psychological symptoms by palliativists. Notwithstanding this should be the gold standard, we know that this is not always possible due to organizational, economic and other constraints. So it happens that patients are referred to the palliative units with a priority established on the basis of their symptoms severity. I think more clarity is required regarding the role that results from the present study may have in this context.

I think the major issues of this Manuscript are associated to the Material and Methods section. Please, address the following points:

3. As first, I would recommend to review the subsections’ titles and/or organization. For example, the subsection 2.2. “Questionnaires used routinely by nursing staff” includes also the description of the questionnaire used as ePROM. At the same time, the procedure for the ePROM administration is described in the subsection 2.1. “Study protocol, patient recruitment”.

4. I would also recommend a more detailed description of the procedure. For example, is the epaAC 2.1 completed based on what the nurse observes and assess during his/her visits? For example, if s/he sees that the patient has not eaten, s/he reports a low score on the item “nutrition”? How long it took for patients to complete the ePOS? Were they instructed to complete it during the same day or could they complete it in different moments? From the Manuscript, it seems this last one is the case, but is it methodologically correct to complete such an assessment in different days, considering that the physical and psychological health may change during the hospitalization?

5. Why do Authors administer a questionnaire (GAD7) and its abbreviated form (GAD2) as well?

6. It should be stated with more clarity which single items have been compared between the two assessment methods.

7. Could have some information been lost when downscaling 4-point Likert scale to a nominal, binary scale (yes/no)?

8. Table 2 reports the subjective need of psycho-oncological or palliative care treatment as assessed by the ePOS, so I do not think this information should be reported here. The totals of ECOG and Metastasized are not 230. Moreover, it is not very cleared to me what Authors mean when they say “due to the fact, that tumor diagnoses, which integrate an unchangeable feature, were 212 counted only once during data collection, the number of patients in this category is less 213 than 230 patients”. Please, clarify.

9. From my understanding of the Manuscript, the nurse-reported data have been collected daily, while the ePROM have been administered once (but some patients started to fill it in 2 or 3 times). If my understanding is correct, I think this should be explained better. Which data from the nurse-reported pool has been used for the analyses? An average of all the collected data (one assessment per day) or only those collected the day in which ePROM has been compiled by the patient? When the una-tantum ePROM has been compiled by the patient? At the beginning of his/her hospital admission, at the end or somewhere in the middle?

10. As Authors say, a large part of their sample did not complete the ePROM. These patients were those with high symptom burden or those that were unfamiliar with electronic devices. However, from Figure 1, it seems that all patients were included in the analysis. Why? Only patients who completed the ePROM should be included in the comparison to avoid biases coming from missing information from underrepresented patients in the ePROM.

11. In general, it seems to me that the nurse-reported assessment is much more oriented to the patient’s physical functioning whereas the ePOS is much more oriented to the patient’s psychosocial health. Is it just my impression? Please, clarify.

Minor editing is required to fix some English errors as well as some typos.

Author Response

Dear Prof. Dr. Mok, dear reviewers,

we sincerely thank you for reviewing the submitted manuscript. Your constructive suggestions improve the significance of our analysis.

We scrupulously dealt with the criticism and the valuable suggestions. During our intense revision we added new information to our sections of introduction, methods, and discussion. Due to the fact, fig 1. does not deliver any new information, we have decided to remove it from our paper.

According to the suggestions that were made, we adjusted the content of our tables.
Besides we aimed to correct our errors regarding the English-language and typos.

All changes are visible in the correction mode. Please find the comments of reviewer #1 and #2 with corresponding changes and explanations in the following table.

We hope that the revision in its current version can be provided for publication in cancers.

With best regards,

Mitra Tewes

Reviewer 2 Report

Following are the comments:

1. Repitation of  advanced cancer patients (advanced cancer patients, ACP) in line 47.

2. line 49, multiple 49 symptoms?? what symptoms discuss here?

3. Line 51, f these professionals, discuss the professionals categorization?

4. What are the differences between PROM and clinician-reported symptom burden? discuss?

5. Why your study is important? discuss plz

6. In method, mentioned the name of center?

7. Patient sampling procedure is not mentioned? plz mentioned how you calculate the sample? and smapling technique?

8. There is no need of table 1 or it will be more clarified?

9. Figure 1 is not readable?

10. Discussion is very brief. Suggested to incorporate more studies for comparison. 

11. Table 4 may be discussed in discussion rather than in conclusion portion.

12. Improve the flow of sentences.

Needs moderate editing and improve the flow

Author Response

(The authors gave the same response as above.)

Round 2

Reviewer 1 Report

I thank the Authors for having addressed my comments and clarified my doubts. I'm satisfied with their responses.

I just recommend one last reading before submission to fix some errors or typo that probably have not been noticed due to the Track Change mode on.

Reviewer 2 Report

I have gone through the paper and its look good. All suggested comments are well incorporated and response.